# Hungry for Sex: Differential Roles for *Ustilago maydis*
*b* Locus Components in Haploid Cells vis à vis Nutritional Availability

**DOI:** 10.3390/jof7020135

**Published:** 2021-02-12

**Authors:** R. Margaret Wallen, Kirsten Richardson, Madison Furnish, Hector Mendoza, Allison Dentinger, Sunita Khanal, Michael H. Perlin

**Affiliations:** Department of Biology, Program on Disease Evolution, University of Louisville, Louisville, KY 40292, USA; r.margaret.wallen@gmail.com (R.M.W.); kirstenmarierichardson@gmail.com (K.R.); MADISON.FURNISH@CUANSCHUTZ.EDU (M.F.); hector.mendoza@louisville.edu (H.M.); allison.dentinger@louisville.edu (A.D.); sunita.khanal@louisville.edu (S.K.)

**Keywords:** mating-type loci, ammonium transceptors, *Ustilago maydis*, fungal dimorphism, host-pathogen interactions

## Abstract

Mating-types allow single-celled eukaryotic organisms to distinguish self from non-self in preparation for sexual reproduction. The components of mating-type loci provide initial self/non-self-recognition through pheromone and receptor interactions that control early cell fusion events. However, they may also provide a second level of scrutiny that requires differences in alleles leading to production of a transcription factor required for successful downstream developmental pathways after initial cell fusion. Interestingly, the protein subunits of these transcription factors have not been thoroughly examined for their roles, if any, in the haploid cells themselves. In *Ustilago maydis*, the causative agent of galls in maize plants, the *b* locus, encoding bEast (bE) and bWest (bW), components of the eventual requisite transcription factor, has been extensively studied for its role in formation of the stable dikaryon after mating and subsequent pathogenic program. Little is known, however, about any roles for bE or bW in haploid cells. Since mating in fungi is often induced under conditions of nitrogen starvation, we have explored connections between the *b* locus and the nitrogen-sensing and response pathways in *U. maydis*. We previously identified a connection in haploid cells between the *b* locus and Ump2, the high-affinity transceptor, a protein that both transports ammonium and triggers filamentous growth as a response to nitrogen starvation. Deletion of the entire *b* locus abrogates the filamentous response to low ammonium, a phenotype that is rescued by overexpression of Ump2. Here we further investigated the individual roles of bE and bW in haploid cells. We show that bE and bW are expressed differentially in haploid cells starved for ammonium. Their respective deletion elicits different effects on transcription of mating and pathogenic-related genes and, importantly, on the degree of pathogenic development in host plants. This is the first demonstration of a role for these mating locus components on haploid development and the first to demonstrate a connection to the ammonium transceptors.

## 1. Introduction

The evolution of sex chromosomes has been widely studied in higher eukaryotes, in exploration of the origin of sexual reproduction and the subsequent creation of genetic diversity. Fungi provide a unique system for the study of the genetic basis for determination of different mating backgrounds required for sexual reproduction [1]. Many fungal species have both haploid and diploid lifecycles, and unlike higher eukaryotes, such as mammals, they have alternatives to sexual reproduction.

The existence of transcription factors directing developmental programs within the sex determining chromosomal regions is found as early as single-celled eukaryotes such as the fungus, *Saccharomyces cerevisiae*, and other yeasts. Sex determination by transcriptional activation finds its evolutionary antecedents in these early eukaryotes, evidenced by the homology between the SRY region of the Y chromosome and the Mc mating-type protein found in the fission yeast, *Schizosaccharomyces pombe* [2]. Although yeast do not possess sex chromosomes, the region of their genome dedicated to determination of mating type, the mating type (*mat* or *MAT*) locus, encodes transcription factors that serve to increase mating type specific gene expression and to repress gene expression of the opposite mating type [3,4,5]. In *S. cerevisiae*, the *MATα* locus produces two transcription factors, one of which (α1) directs expression of α-specific genes and the other of which, α2, suppresses expression of a-specific genes [3,4]. As with human sex chromosomes, other mating type specific genes are found outside of the MAT locus, including those encoding pheromones and pheromone receptors that are necessary to elicit the mating response [3,4]. Additionally, of note, as with human sex determination, *S. cerevisiae* has a default program (mating type a) that can be over-ridden by the presence of a dominant mating type determination actor produced by the MATα locus.

At the other end of the fungal spectrum of mating type loci is *Microbotryum lychnidis-dioicae*. This smut species contains the first identified sex chromosomes in a fungus, with genes for the homeodomain transcription factors, pheromone precursors, and pheromone receptors all located within a single genetic region on a specific chromosome in each mating-type haploid strain [6]. Like their human counterparts, these fungal mating type determining chromosomes contain pseudoautosomal regions, indicating that they may have arisen from autosomes and therefore genetic regions that had other functions not related to mating [7]. Additionally, like in the human system, these homeodomain transcription factors are responsible for regulating the expression of genes outside of the mating type chromosomes [6].

As in some cases, mating type determination is the result of the interplay of activation of transcription factors within the mating type loci and their regulatory targets outside of the mating type locus, we sought to understand the role these transcriptional activators may have played before the evolution of sexual reproduction. Our model organism is *Ustilago maydis*, a basidiomycete pathogen of maize. In *U. maydis*, there are two regions of the genome located on different chromosomes that determine mating type. The *a* mating type locus of *U. maydis* encodes a pheromone precursor and pheromone receptor to the pheromone produced by the opposite mating type. The *b* locus codes for two homeodomain proteins, bEast (bE) and bWest (bW). During mating, products of the *b* locus originating from opposite mating types dimerize to form a functional transcription factor necessary for infection and proliferation within the host plant [8].

In the canonical mating pathway, the interaction of a pheromone from one mating type with a pheromone receptor on a *U. maydis* cell of the opposite mating type results in phosphorylation of Prf1, the master transcription factor for mating activation [9]. Prf1 binds to response elements in both the *a* and *b* mating type loci, leading to increased transcription of both [9]. The interaction of pheromone with pheromone receptor also results in the growth of conjugation tubes and eventual fusing of haploid cells. Within the resultant dikaryon, if the two nuclei have different alleles at the *b* mating type loci, the heterodimer forms and transcription is activated to produce proteins necessary for pathogenicity [10]. As the genetic regions are located in the haploid cells prior to fusion and the subsequent dikaryotic/diploid portions of the *U. maydis* lifecycle, we wondered what role, if any, these particular genetic sequences and products might have in the haploid cell. Specifically, we hypothesized that the components of the *b* mating type locus, in particular, may play a role in response of haploid cells to nitrogen limiting conditions.

Several conclusions from previous work in the lab prompted further investigation of the interplay between mating type loci components and nitrogen response components in *U. maydis*. First, deletion of either the *b* mating type locus or the *ump2* gene for high affinity ammonium transporter/sensor (“transceptor”) resulted in an abnormal haploid phenotype when grown in nitrogen limiting conditions. This was demonstrated in both FB1 and FB2 strains (harboring different alleles at both the *b* and *a* mating-type loci). Second, while deletion of the *b* mating type loci would obviously preclude successful mating with a compatible strain, surprisingly we found that deletion of the *ump2* loci in potential partners also resulted in a similar loss-of-function [11]. Third, components of the *b* mating-type loci in the FB1 background are differentially regulated under Low-NH_4_ conditions. While nitrogen-limitation is a prerequisite for mating in a lab setting for this fungus and a variety of others [12], strains in the lab are often propagated separately, such that mating in response to resource scarcity is not possible. Due to this relationship between nitrogen availability and gene expression from the *b* mating type loci [10], we sought to understand what role the individual bE and bW components might play in the phenotype and behavior of haploid strains and if these behaviors were general or varied depending on the composition of the *b* mating type loci.

## 2. Materials and Methods

### 2.1. Strains and Growth Conditions

For analysis of mating-related targets on charcoal plates, *U. maydis* cells were grown at 30 °C on either Potato Dextrose Agar (PDA, Difco™, VWR Scientific, Radnor, PA, USA) or PDA supplemented with 1% activated charcoal (J.T. Baker, VWR Scientific, Radnor, PA, USA). Haploid or mated cells were typically incubated for 48 h.

*U. maydis* cells were grown at 30 °C on Array Medium [AM] (6.25% Holliday Salt Solution [13], 1% glucose, 30 mM Glutamine/50 mM Ammonium sulfate and 2% agar) (High NH_4_) and Array Medium [AM] with low ammonium (Low NH_4_) (6.25% Holliday Salt Solution [13], 1% glucose, 50 µM ammonium sulfate and 2% agar) for 48 hrs. In order to assure that any filamentation observed on the latter medium was not due to acidification of the media, the strains were also tested on AM with Low NH_4_ that had been buffered with Tris-HCl, pH 7.0; filamentation was comparable in all cases to what was observed on unbuffered AM Low NH_4_ plates. *U. maydis* strains used are listed in Table 1. All mutants were generated in either the FB1 or FB2 backgrounds [10].

### 2.2. Genetic Manipulation and Vector Construction

Mutants having deletion or overexpression of *ump2* in *U. maydis* were obtained by homologous recombination as described previously [16]. The *ump2* deletion construct was created using a 6 kb PCR product from strain um2h-2, Δ*ump2 a1b1* [17] generated using primers Ump2KOup5′ (GGCAAGACAAGACGAGAAGA) and Ump2_Dn_TapR (TGCGTGTCTCAAACTCCTCT). The *ump2* overexpression construct was produced by amplifying the *ump2* ORF from the plasmid pUmp2 303 using the Ump2_CGO5′ (ATTAACCGCGGAAATGGTTAACGCCAGCTAC) and Ump2_CGO3′ (TGATTGCGGCCGCTTAGACAGCAGTAGGCTG) primers and cloning the product into pCR2.1 TOPO (Invitrogen) [17]. To provide constitutive expression, the *ump2* ORF was cloned after the P*otef* promoter between the *SacII* and *NotI* sites of the p123 vector (Otef expression vector) [18]. The Otef expression vector was linearized using the restriction enzyme *Ssp1* before transforming *U. maydis* to select for recombinants at the *ip* locus, providing carboxin resistance [19]. The construct for making the *b* gene deletion in *U. maydis* strain FB1 (*a1 b1*) was obtained from Dr. J. Kämper J [20] Complementation of *ump2* similarly used the ORF cloned with or without its native promoter into the Otef expression vector. Complementation of the *b* deletion mutant was achieved by cloning both ORFs (and promoter regions) from the wild-type *b* mating locus into the Otef expression vector. This construct was introduced into *b* deletion mutants as previously described [11].

### 2.3. RNA Isolation and Expression Analysis

*U. maydis* cells were grown on AM-glutamine high ammonium (High NH_4_) and AM-low ammonium (Low NH_4_) for High- vs. Low-Ammonium Analysis and PDA or PDA-charcoal plates for 48 h. RNA isolation for the transcriptional profile was as described previously [11,21]. These cells were further processed using the RNeasy plant mini kit (Qiagen, Valencia, CA, USA) or the Quick-RNA™ MiniPrep Plus (Zymo Research, Irvine, CA, USA) following the manufacturer’s instructions. The integrity of the RNA samples was checked using an Agilent Bioanalyser 2100 (Agilent, Santa Clara, CA, USA). The RNA samples were treated with DNaseI (New England Biolabs, Ipswich, MA, USA) before proceeding to the preparation of the cDNA samples. The double stranded cDNA samples were prepared with Super Script III (Invitrogen, Carlsbad, CA, USA) following manufacturer’s instructions.

Primers for the different genes (Table 2) were designed using the ABI Primer Express software version 3.0, ensuring that all the primer sets investigated had the same amplification efficiency. The generated cDNA was diluted 10-fold. All real-time PCR reactions were performed in a 62.4 µL mixture containing 1/10 volume of the diluted cDNA preparation, 1X Power SYBR green PCR master mix (Applied Biosystems, Foster City, CA, USA), 1.6 µL each of the 2.5 µM forward and reverse primers, making up the reaction volume to 62.5 µL with dH_2_O. Quantifications of RNA expression levels were performed in an Applied Biosystems Step-One thermocycler using the following PCR conditions: 95 °C for 10 min followed by 95 °C for 15 secs and 60 °C for 1 min for 40 cycles. Melting curve analysis was performed at the end of each cycle to ensure specificity of the reaction. The concentration was determined by the comparative CT method (threshold cycle number at the cross point between amplification plot and threshold) and values were normalized to expression of the constitutively expressed gene, *eif2B*, encoding the translation initiation factor eIF2. Changes in the gene expression are averages of at least three biological replicates and are displayed as log2-fold changes relative to expression of wild-type FB1 under High-NH_4_ condition or as a comparison of the mutant at High and Low NH_4_. Target gene expression was normalized to FB1 WT. For PDA and PDA-charcoal analysis, target gene expression was normalized to FB1 WT on PDA. For High vs. Low NH_4_ analysis, target gene expression was compared to FB1 WT on High-NH_4_ medium. Normalization was done using the Pfaffl Method [22]. All of the analyses were done using a one-way ANOVA with a Dunnett’s multiple comparison post hoc test.

### 2.4. Mating Assay and Plant Pathogenesis

Cell densities of liquid cultures were measured spectrophotometrically. Liquid overnight cultures were diluted in fresh media to obtain an Absorbance at 600 nm (A600) of 0.1. The newly inoculated culture was allowed to grow for an additional 4 h to obtain an A600 between 0.5–0.7 (exponential growth phase). Mating assays were performed using 10^7^ cells/mL and spotting 10 µL onto PDA charcoal as previously described [15,23]. Plant infection using 8–10-day-old Golden Bantam corn seedlings (Bunton Seed Co., Louisville, KY, USA and W. Atlee Burpee & Co., Warminster, PA, USA) was performed with a cell density of 10^8^ cells/mL for pre-mixed haploid strains of opposite mating type as previously described [23] and virulence was rated by a disease index (DI) on a scale of 0 to 5 [14]. Statistical analysis of the disease index measures was performed using a Kruskal–Wallis Test with a Multiple Comparison Test in R [24].

## 3. Results

### 3.1. Under Mating Conditions, FB1 and FB2 Do Not Respond in the Same Way at the Level of Regulation of Gene Expression

The standard assay to determine if mating is affected by a particular mutation in a strain of *U. maydis* is the charcoal “fuzz” phenotype [23]. Two strains that are able to form a dikaryon will produce a white-colored “fuzz” on the charcoal plate, the result of aerial hyphae formation. Similarly, a haploid solopathogenic strain [15] or a stable diploid strain [8] will also produce a positive fuzz response on charcoal media. This charcoal mating media is usually produced by supplementing growth media with 1% activated charcoal. Activated charcoal used in this assay is known to sequester ammonium, much as it does in filtration systems, resulting in the formation of a nitrogen-limited environment [25]. Our first step in understanding how the components of the *b* mating type locus contribute to the Low-NH_4_ response in haploid cells was to examine how these haploid strains behaved when plated in the absence of a mating partner. Additionally, the mated pair (FB1 and FB2) was used as a comparison to determine if the contribution of the two distinct protein products (i.e., bE and bW) of each *b* mating type allele was equal.

One of the media typically used in this mating assay is Potato Dextrose Agar (PDA) which is by no means nutrient-replete. It is produced from a potato infusion, and though many fungi are able to grow well on this undefined medium, it is lower in nutrients compared to other richer, commonly used media, such as YEPS or complete medium [13]. In order to take into consideration the nutrient limitation already posed by the PDA, we used haploid strains grown on PDA without charcoal as a baseline for comparison. Both FB1 and FB2 strains showed a similar phenotype on low-ammonium media, including PDA, formation of narrow hyphae that extended out onto the surrounding media (Figure 1A). However, they did not respond with the same differential gene expression. On AM Low-NH_4_ medium, FB1 shows a dramatic increase in *ump2* expression as compared with the same strain on AM High NH_4_ ([11]; also, see Appendix A). While FB1 increased *ump2* expression on PDA-charcoal (*p* < 0.0.05), the magnitude of this increase was far less than what was observed comparing expression on AM Low NH_4_ with AM High NH_4_ [11], suggesting that the difference in available ammonium concentrations was not as great in PDA vs. PDA-charcoal. Moreover, the expression level remained unchanged for FB2 on media containing charcoal as compared to PDA alone. Of note, the mated pair also showed increased *ump2* expression as compared to the same mixture of both strains on PDA without charcoal (*p* < 0.001) (Figure 1B). As there was no significant difference in the expression of *ump2* in FB2 relative to the expression of this ammonium transceptor in FB1 when both haploid strains were grown on PDA alone, the upregulation of the mated pair on PDA-charcoal was not simply the additive effect of the two haploids. In addition to the mating loci, FB1 and FB2 are not isogenic, but they are congenic [24]. As the morphological response of both FB1 and FB2 was the same under Low-NH_4_ conditions and the deletion of *ump2* interrupts this response, it was striking that the change in *ump2* transcript level on PDA to the relatively lower ammonium condition (i.e., plus charcoal) was not the same in both strains.

### 3.2. The Components of the b Mating Type Locus Are Not Expressed Equally

Before we limited our investigations to haploid strains alone, we further analyzed the contribution of different alleles of the *b* mating type locus during mating-inducing conditions. As described above for our examination of *ump2* expression level, these comparisons were also done by examining each haploid strain on PDA as compared to PDA supplemented with 1% activated charcoal. The level of expression from each allele was also compared to expression when compatible haploid strains were mixed together. The two products of the FB1 locus, *bE1* and *bW1*, were both upregulated in the mated pair when plated on charcoal PDA (*p* < 0.01) (Figure 1C,D), while neither of the products of the FB2 locus, *bE2* or *bW2*, showed any significant variation in expression level across the strains and conditions examined (Figure 1E,F). Again, the magnitude of increases in expression from High- to Low-NH_4_ conditions was substantially lower for both *bE1* and *bW1*, as compared to FB1 grown on AM medium ([10]; Appendix A). As the genes of the *b1* mating type locus, *bE1* and *bW1*, seem to be expressed differentially to a greater extent than the homologous components of the *b2* mating type locus, we chose to focus on the FB1 strain to better understand the relationship between Low-NH_4_ response and mating type locus.

### 3.3. Overexpression of ump2 in Haploid FB1 Lacking the Entire b Mating Type Locus Rescues the Loss of Filamentous Phenotype in Response to Low NH_4_

The deletion of either *ump2* or the *b* mating locus in the FB1 background results in the loss of typical filamentation when the haploid strain is grown on ammonium-depleted medium [11]. Additionally, overexpressing *ump2* in the FB1 wild-type (WT) background produces this same filamentous behavior even in High-NH_4_ conditions [11]. To further elucidate the role of the *b* mating type loci in the response of the haploid strain to Low-NH_4_ conditions, we generated a strain that constitutively expressed *ump2* in a genetic background deleted for the entire *b* mating locus (FB1Δ*b ump2^Potef^*). While this double mutant strain lacked the hyper-filamentation phenotype observed when *ump2* is constitutively expressed in the FB1 background, in Low-NH_4_ media, FB1Δ*b ump2^Potef^* produced filaments that were indistinguishable from those produced by the FB1 WT haploid strain grown on similar medium (Figure 2A).

To ensure that the phenotypes observed with genetic manipulations of the FB1 background were not artifacts of some other characteristic of the strain, we also investigated these phenotypes in 1/2, another *U. maydis* strain that carries the same alleles at the *b* (and *a*) mating locus [14]. As we had observed with FB1, deletion of the entire *b* mating type locus resulted in the loss of filamentation under Low-NH_4_ conditions and constitutive expression of *ump2* rescued this loss of function (Figure 2B).

Not surprisingly, the overexpression of the high affinity ammonium transporter did not rescue the loss of mating ability in the *b* mutant as assessed by charcoal mating assay (Figure 2C). All mated pairs including the FB1Δ*b ump2^Potef^* strain had an odd phenotype, potentially as a result of the haploid filamentous phenotype, but none demonstrated the white “fuzz” phenotype characteristic of a compatible mating pair. The production of aerial hyphae by compatible mating partners requires the formation of a heterodimer between products of different *b* alleles [26,27]. Overexpression of *ump2* was only able to rescue loss of ammonium response phenotypes in haploid strains, not those related to mating.

### 3.4. Haploid Strains Have Different Phenotypes under Low-NH_4_ Conditions When Only Part of the b Mating Type Locus Is Deleted

The roles of *bE* and *bW* alleles in mating and pathogenicity have been extensively examined [15,28]. We reconfirmed that partial *b* deletion mutants behaved typically when mated with a compatible partner. Both *bE1* and *bW1* mutants produced the white “fuzz” phenotype when spotted with a compatible WT mating partner on charcoal plates (Appendix A). Moreover, both also were able to form aerial hyphae when spotted with mating partners that only possessed part of the compatible *b* mating locus (for example, a strain bearing *bE1* could mate with a strain bearing *bW2*) (Appendix A). This was not unexpected as it has long been the paradigm in *U. maydis* mating that a single *b* heterodimer is sufficient to produce virulence and upregulation of mating-related targets [28], although it was recently also determined that not in all cases are both of the possible bE and bW combinations functional [1]. Similarly, both partial *b* mutants were able to establish infection comparable to a WT mating when inoculated into host corn seedlings with a compatible mating partner.

Investigations involving gene expression from the *b* mating type locus during nutrient-limited conditions (i.e., AM medium with Low NH_4_) revealed that *bE1* and *bW1* were not expressed at the same levels in haploid strains [11], prompting the hypothesis that potentially these two products of the *b* mating type locus had different functions in the haploid. Our initial experiments focused on observing phenotypic changes as a result of partial deletion of the *b* mating type locus under Low-NH_4_ conditions. Although the phenotype of *U. maydis* colonies with either *bE1* or *bW1* deleted did not show the drastic impairment of filamentation observed when the complete *b* locus was removed (Figure 3A), they did show decreased filamentous behavior as compared to the FB1 WT (Figure 2A). The greatest reduction of filamentation was observed when *bE1* was deleted, while deletion of *bW1* resulted in shorter filaments than the WT strain when grown on the same medium. As with the FB1 WT parental strain, phenotypic changes were only observed when ammonium supply was limited, and colonies grew normally on High-NH_4_ media (Figure 3A).

As overexpression of *ump2* in a strain with the entire *b* mating type locus deleted resulted in rescue of the loss of filamentous phenotype on Low-NH_4_ media, we also investigated whether a similar phenomenon would occur in these partial *b* deletion strains. When only *bE1* was deleted, overexpression of *ump2* resulted in a phenotype similar to both the WT under Low-NH_4_ conditions and of the complete *b* deletion mutant constitutively expressing the high affinity ammonium transporter. However, when *bW1* was missing, overexpression of *ump2* resulted in an exaggeration of the aberrant filaments in the parental strain lacking *bW1* (Figure 3B).

### 3.5. Regulation of Gene Expression Is Different in Partial b Mutants Overexpressing ump2

The construct used to drive the exogenous expression of *ump2* included the constitutive *otef* promoter, and theoretically, the expression of *ump2* from this particular locus should be approximately the same in all transformed mutants; however, we observed variation in the expression levels of *ump2* within these mutant strains that could not be accounted for by relatively constant exogenous expression and therefore could be attributed to changes in endogenous *ump2* expression. Although *ump2* expression is indispensable for the filamentation phenotype, high levels of expression under Low-NH_4_ conditions are not sufficient for filamentation, as the complete *b* deletion mutation had *ump2* levels comparable to those of FB1 WT under Low-NH_4_ conditions, but failed to filament (Figure 3C). Both partial *b* mutants showed impaired expression levels of *ump2* under Low-NH_4_ conditions (i.e., reduced up-regulation relative to that of the same strain on High NH_4_), potentially accounting for their reduced, although not abolished, ability to produce filaments when grown on Low-NH_4_ media (Figure 3C). FB1Δ*bW1ump2^Potef^* showed a substantially higher level of *ump2* expression than any other strain, including FB1 WT, on the filament-inducing media (Figure 3C). As the change in *ump2* expression was more variable than could be accounted for by constitutive expression from the *otef* promoter alone, this suggested possible interactions with the endogenous *ump2* promoter, either via Ump2 levels directly, or via interplay with other genes. Thus, we sought to understand at the transcription level how various mutants responded to different combinations of *ump2* and *b* locus component expression.

Our previous investigation of the interplay between *ump2* expression, *b* mating loci expression, and haploid phenotypes had led to the discovery that alteration in the expression levels of these two regulators could have downstream effects on the expression levels of other targets that previously have been considered functional only during mating, including *mfa1.* The protein product of the *mfa1* gene is a pheromone precursor [27]. In FB1 WT, under Low-NH_4_ conditions, *mfa1* expression increases, producing more pheromone, potentially as a way of increasing the likelihood of attracting a compatible mating partner. This finding was confirmed for both *mfa1* and *mfa2*, for strains 1/2 and 2/9, respectively (Appendix A). In the canonical mating pathway, the interaction of pheromone with pheromone receptor results in a cascade of events leading to the phosphorylation of Prf1 [9]. The Prf1 transcription factor, when appropriately phosphorylated, increases expression of the *bE* and *bW* genes [9]. However, in the haploid, it appears that the presence of an intact *b* mating type locus is required for normal *mfa1* expression. Under High-NH_4_ conditions, complete deletion of the *b* locus had little effect on the expression level of *mfa1*; however, in both *bW1* and *bE1* mutants, *mfa1* expression was elevated even in these replete conditions. Under Low-NH_4_ conditions, a sharp upregulation of *mfa1* was observed in the wild type FB1. Neither partial *b* mutant nor the FB1 complete *b* deletion mutant showed a dramatic change in *mfa1* expression on nutrient-depleted media, having levels approximately the same as the respective strain on High-NH_4_ media (Figure 3D). However, the bW1 deletion strain did show a significant increase in mfa1 expression on Low-NH_4_ medium, compared with FB1 wild type on nutrient-replete medium (Figure 3D). Interestingly, constitutive ectopic expression of *ump2* increased *mfa1* expression in the complete *b* deletion mutant and in the partial *bE1* deletion mutant (but not for FB1Δ*bW1*), suggesting that *bE1* and *bW1* have different and non-redundant roles in the haploid cell (Figure 3D). Expression level of *mfa1* alone was not sufficient to account for filamentous phenotype, as the *bE1* mutant overexpressing *ump2* showed elevated levels of *mfa1* on High-NH_4_ media but failed to filament (Figure 3B,D). Other traditionally mating-related targets were investigated in the FB1 background, such as expression of the *pra1* gene, encoding the pheromone receptor, *prf1*, the gene for the mating transcription factor, and *rbf1*, the gene for the master regulator of *b* heterodimer response, to determine if expression of these targets also changed with different levels of *ump2* expression and different components of the *b* mating type locus intact. For wild type FB1, *prf1* and *rbf1* expression each increased significantly on Low NH_4_ (results that were reproduced in the 1/2 strain; see Appendix A). Deletion of the entire *b* locus, was associated with substantial increase in *pra1* expression (Figure 4A). However, either complete or partial deletion of the *b* mating type locus resulted in decreased expression of *prf1* and *rbf1* under Low-NH_4_ conditions. The only other notable changes were a significant (*p* < 0.05) increase of *pra1* expression in the *bE1* deletion mutant on High NH_4_ and on both High- and Low-NH_4_ in this mutant when simultaneously overexpressing *ump2*; additionally, the *bE1* deletion mutant had a significant (*p* < 0.01) increase in *rbf1* expression on High NH_4_. (Figure 4).

Rbf1, the master regulator of the *b*-responsive changes in transcription, is known to be involved in cell-cycle arrest [28]. While FB1 WT showed a strong upregulation of *rbf1* under Low-NH_4_ conditions (as compared to the same strain on High-NH_4_ media), an increase of similar magnitude was not observed in the complete *b* deletion mutant, partial *b* deletion mutants, or either of the previous two overexpressing *ump2.* In fact, deletion of *bE1* actually resulted in higher expression of *rbf1* on High-NH_4_ media.

Importantly, taken together, this is the first evidence of three major findings in the role of mating type loci in haploid fungal cells: (1) the components are necessary for typical haploid behavior (i.e., filamentation under Low-NH_4_ conditions), (2) the roles of *bE1* and *bW1* in the haploid cell are different, and (3) other proteins generally believed to be involved only post mating, such as Rbf1, may also have roles in haploid cells. Since functional Rbf1 need only be present in one of the mating partners for successful mating [29] these observed differences in *rbf1* transcription here suggest additional functions in such haploid cells.

## 4. Discussion

The role of the heterodimer produced by the interaction of proteins from different alleles of the *b* mating type locus is well characterized. Although few direct targets of the heterodimer have been found, the bE/bW complex is responsible for upregulating *rbf1*, a transcription factor normally required for directing gene expression leading to cell cycle arrest and pathogenicity [29]. Here, we provide evidence that the *b* mating type locus also plays a role in the normal phenotype of haploid cells, particularly with respect to Low-NH_4_ response. The requirement in mating for nitrogen starvation or Low NH_4_ is well established for a range of organisms, e.g., algae [30] and fungi [12,31] including *U. maydis* [32,33,34] The expression of *bE1* (and potentially *bW1*) is increased under Low-NH_4_ conditions, as well as nitrogen-limiting conditions found in PDA-charcoal agar used for typical mating assays. Partial deletion of the *b* locus in the FB1 background results in an atypical haploid phenotype, specifically loss of filamentous behavior when grown on media containing minimal concentrations of ammonium. Our results indicate the potential for other canonically mating-related proteins to have functions in haploid cells. Further investigation into the cell cycle status of these haploid mutants would be needed to determine if changes in expression levels of *rbf1* had implications on cell cycle arrest.

To corroborate our assertion that the *b* mating type locus is involved in the nitrogen availability response of the haploid cell, the reduced filamentation in the partial *b* deletion strains could be rescued by overexpressing *ump2*, the high affinity ammonium transporter, in these backgrounds. Ump2 is known to be upregulated when haploid cells are exposed to Low-NH_4_ media and during biotrophic growth stages in maize [34]. While there was slight upregulation of *ump2* under Low-NH_4_ conditions in both of the partial *b* deletion mutants, the magnitude of increased expression was much higher in the complete *b* deletion mutant and was indistinguishable from FB1 WT. That said, deletion of the entire *b* mating locus resulted in a more severe loss of filamentous behavior of haploid cells under these conditions, suggesting that increased *ump2* expression alone was not responsible for the filamentous phenotype of WT on Low NH_4_. The strong up-regulation of *ump2* in the strain bearing only bE1 while additionally over-expressing *ump2* from the ectopic *otef* promoter, suggests some sort of interaction between bE1 and Ump2 that facilitates such increased transcription. That this is not directly due to an effect on the endogenous *ump2* promoter is suggested by experiments where the Ump2 protein expressed from that promoter has been tagged with YFP and compared in FB1 and FB1 over-expressing *ump2* from the ectopic *otef* promoter (Appendix A). In this case, the level of YFP fluorescence does not obviously increase due to the over-expression ectopically of untagged Ump2. However, this does not address possible effects on Ump2 stability when overexpressed from the ectopic promoter.

Upon entry into the host plant, one of the first changes in plant gene expression is related to creating pools of nitrogen sources that the fungus can use [34]. This, along with the practice of using nitrogen limited charcoal media to detect mating, suggests that the lack of available nitrogen is a stimulating factor for mating behavior. In haploid cells, when experiencing limited ammonium, an upregulation of the pheromone precursor, Mfa1, is also observed, as the haploid is potentially trying to increase its probability of attracting a mate. In the absence of the entire *b* locus, i.e., absent bE1 and bW1, this upregulation fails to occur, suggesting that in the haploid cell, the *b* mating type locus may be responsible for behavior that prepares the cell for mating. In the complete *b* deletion strains, overexpressing *ump2* increases *mfa1* expression under Low-NH_4_ conditions. Yet, such a strain is still unable to mate with its wild type FB2 partner. Thus, despite the rescue of filamentation on Low NH_4_, and the increased *mfa1* expression, as expected, overexpression of *ump2* in in this background failed to rescue the mating defect associated with deletion of the entire *b* locus. By comparison, in partial *b* deletions, only when bW1 is present does overexpression of *ump2* result in a similar induction in pheromone precursor gene expression. When *bE1* was deleted, the expression level remained unchanged from High to Low-NH_4_ conditions. With *ump2* overexpression, this relatively constant level of *mfa1* expression was also observed, although the magnitude of expression of *mfa1* was higher in FB1Δ*bE1ump2^pOtef^*. These results together suggest that *bE1* and *bW1* have different roles in haploid cells and both interact somehow at the transcriptional level with *ump2* regulation.

While the observations in this report are intriguing, they do not yet provide a mechanism whereby alleles at the *b* locus interact with the Ump2 transceptor directly or via transcription of its gene. We are currently undertaking RNA-Seq experiments to begin to identify *b* targets in haploid cells. Identification of such targets will be an important step in understanding the role of bE and bW in haploid sporidia. Further, similar experiments where Ump2 is overexpressed in *b* complete or partial deletion strains, will likely better explain the role of Ump2 in this interaction between the two loci. Additionally, we have not here measured levels of expression of these target genes at the protein level, which may provide further insights as to the level(s) at which these putative interactions take place.

Evidence suggests that bE1 and bW1, from the same *b* mating type locus, are not able to make a functional heterodimer [1,35]. However, it is possible that each acts independently of the other or interacts with other proteins to serve as different transcription factors that may direct gene expression of haploid cells under Low-NH_4_ conditions. Although canonically involved in the mating and pathogenicity pathways, our results suggest a role for these proteins in the haploid lifecycle as well. Here, we observed differences in expression levels for different *bE* and *bW* alleles (Figure 1E,F). We do not understand the basis of this finding, but it would be interesting to examine additional *b* locus alleles for differences in levels of expression in haploids as a response to nitrogen availability. The difference in phenotype observed in strains lacking only part of the *b* mating locus suggests non-redundant, indispensable functions of these proteins in the haploid cell. It is likely that, evolutionarily, a response to depleted resources such as nitrogen predated the sexual lifecycle of *U. maydis*. As mating and entering into the host plant became a strategy for accessing nitrogen pools previously unavailable outside of the host, components of the genome previously involved in nutrient sensing and response became linked to changes in gene expression leading to mating and host entry. To better understand the role of the components of the *b* locus in the haploid cells, further investigation is needed to identify other potential interacting partners and changes in gene regulation.

## Figures and Tables

**Figure 1 jof-07-00135-f001:**
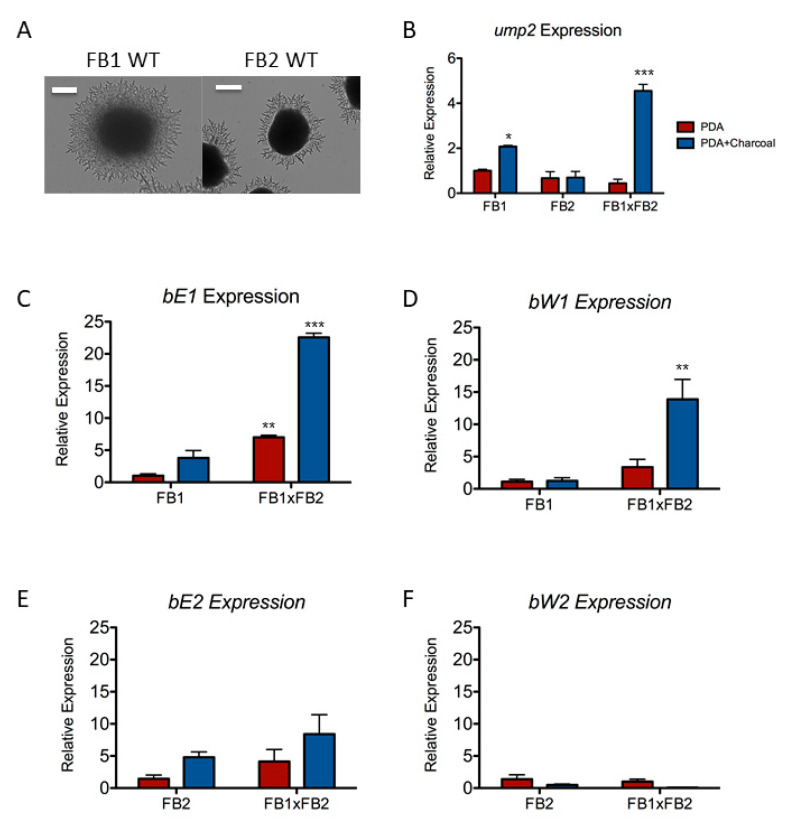
Phenotype of FB1 and FB2 under Low-NH_4_ conditions. (**A**) Filamentous phenotype of FB1 and FB2 when grown on PDA media. Scale bars show, for FB1, 100 µM; for FB2, 200 µM. (**B**) Differential expression of *ump2* on PDA vs. PDA supplemented with 1% activated charcoal. FB1 shows upregulation of *ump2* on charcoal, as does the FB1xFB2 mated pair, while FB2 has similar levels of *ump2* expression on both media. (**C**) *bE1* expression on PDA vs. PDA-charcoal. FB1 expression level of *bE1* shows a trend towards significance (*p* = 0.07) on PDA-charcoal with respect to PDA. *bE1* is significantly upregulated in FB1xFB2 on both PDA and PDA-charcoal. (**D**) *bW1* expression on PDA vs. PDA-charcoal. The expression level of *bW1* is the same on PDA vs. PDA-charcoal. In the FB1xFB2 mated pair, *bW1* expression is significantly increased. (**E**) *bE2* expression on PDA vs. PDA-charcoal. No change in *bE2* expression was observed on either type of media, even in the mated pair FB1xFB2. (**F**) *bW2* expression on PDA vs. PDA-charcoal. A similar trend was observed as with *bE2*, with no difference in expression. For measuring relative levels of expression, the Comparative CT method (threshold cycle number at the cross point between amplification plot and threshold) was used and values were normalized to expression of the constitutively expressed gene, *eif2B*. Normalization was done using the Pfaffl Method [22]. Changes in gene expression are averages of at least three biological replicates and are displayed as log2-fold changes ± S.D., relative to expression of wild-type FB1 on PDA. Relative expression of each strain and/or condition was compared to the reference strain. All of the analyses were done using a one-way ANOVA with a Dunnett’s multiple comparison post hoc test. (* *p* < 0.05, ** *p* < 0.01, and *** *p* < 0.001).

**Figure 2 jof-07-00135-f002:**
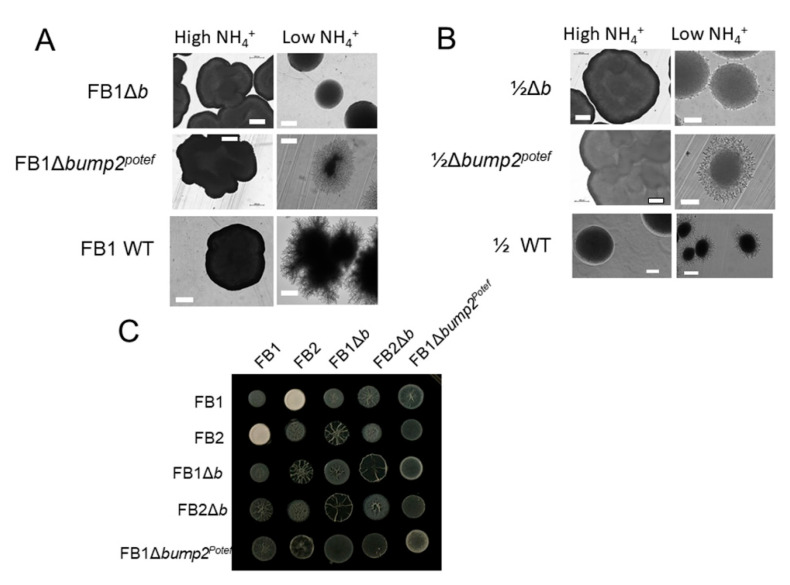
Phenotypes of partial *b* locus deletions with and without *ump2* overexpression. The filamentous phenotype of haploid *U. maydis* colonies is lost when the entire *b* mating type locus is deleted in the FB1 background (**A**) as well as 1/2 background (**B**). Overexpression of *ump2* in both of these backgrounds rescues the loss of filamentous phenotype. (**C**) Although the haploid phenotype is restored by overexpression of *ump2*, the loss of production of aerial hyphae in a charcoal mating assay with a compatible mating partner is not restored. The *ump2* overexpressor in the complete *b* deletion background does have a slightly fuzzy phenotype, potentially as a result of filamentation. High-NH_4_ media contains 50 mM ammonium sulfate, while Low-NH_4_ media contains 50 µM ammonium sulfate. Size bars, (**A**), 200 µm; (**B**), High NH_4_, 200 µm; Low-NH_4_, 100 µm.

**Figure 3 jof-07-00135-f003:**
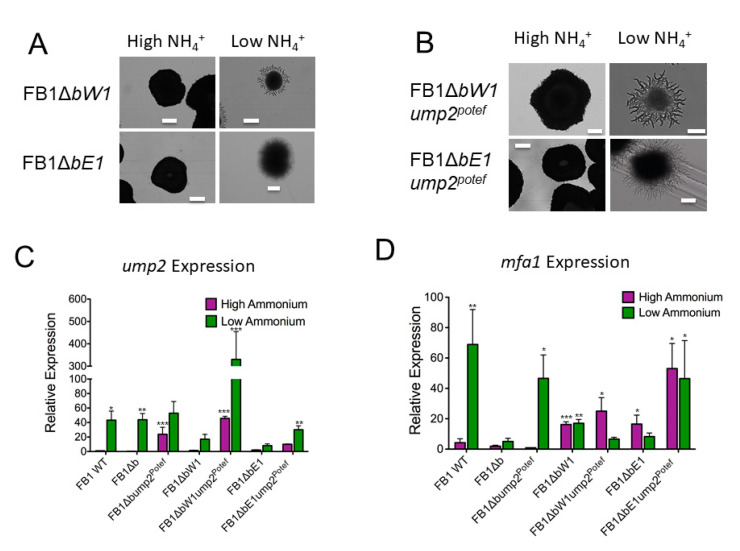
Partial *b* deletion and overexpression of *ump2* in partial *b* deletion backgrounds. (**A**) Both *bW1* and *bE1* deletions show a reduction of filaments under Low-NH_4_ conditions. Size bars, 200 µm. (**B**) Constitutive expression of *ump2* in a *bW1* deleted background results in increased filamentation, although the phenotype was slightly different, with very thick filaments. The overexpression of *ump2* in a *bE1* deleted background resulted in rescue of the visually normal phenotype (i.e., filamentous on Low-NH_4_). Size bars, High NH_4_, 200 µm; Low NH_4_, 100 µm. (**C**) qRT-PCR analysis of *ump2* expression. In FB1 WT, *ump2* is upregulated when the haploid strain is plated on Low-NH_4_ media. A similar trend is observed in the complete *b* deletion strain. Both partial *b* deletion mutants show no significant difference in *ump2* expression as compared to FB1 WT on High NH_4_ media. While *ump2* overexpression markedly increases overall *ump2* transcript level in the *bW1* deleted background, a similar trend is not observed in the *bE1* deletion background. (**D**) qRT-PCR analysis of *mfa1* expression. FB1 WT shows dramatically increased *mfa1* expression under Low-NH_4_ conditions as compared to the same strain on High NH_4_. While this phenotype is abolished in the complete *b* deletion mutant, overexpressing *ump2* increases *mfa1* expression on Low NH_4_ in the *b* deletion background. Deleting *bW1* results in increased levels of *mfa1* on both high and Low NH_4_ as compared to FB1 on High-NH_4_ media. Overexpression of *ump2* in a *bE1* deleted background results in high levels of *mfa1* under both media conditions. High-NH_4_ media contains 50 mM ammonium sulfate, while Low-NH_4_ media contains 50 µM ammonium sulfate. Changes in gene expression are averages of at least three biological replicates and are displayed as log2-fold changes ± S.D., relative to expression of wild-type FB1 on High NH_4_. Relative expression of each strain and/or condition was compared to the reference strain. All of the analyses were done using a one-way ANOVA with a Dunnett’s multiple comparison post hoc test. (* *p* < 0.05, ** *p* < 0.01, and *** *p* < 0.001).

**Figure 4 jof-07-00135-f004:**
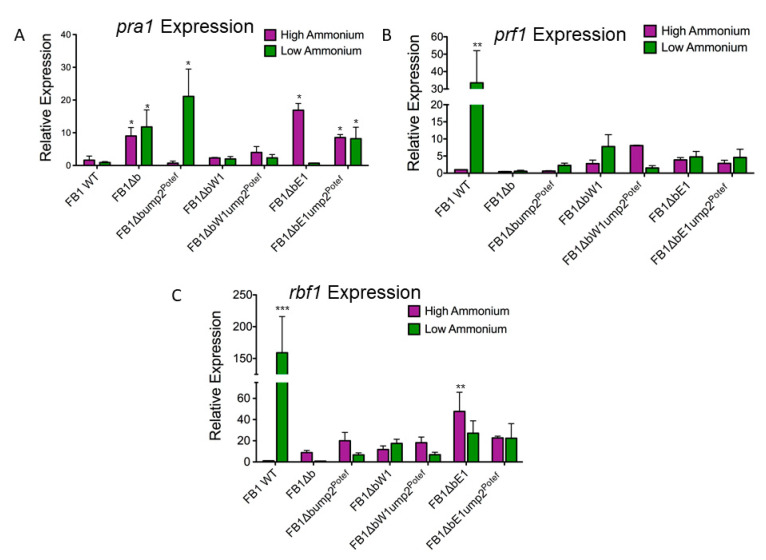
Change in expression of other mating related targets. The transcript level of *pra1* (**A**), *prf1* (**B**), and *rbf1* (**C**) were evaluated in different *b* deletion strains without and without constitutive expression of *ump2*. Changes in gene expression are averages of at least three biological replicates and are displayed as log2-fold changes ± S.D., relative to expression of wild-type FB1 on High NH_4_. Relative expression of each strain and/or condition was compared to the reference strain. All of the analyses were done using a one-way ANOVA with a Dunnett’s multiple comparison post hoc test. (* *p* < 0.05, ** *p* < 0.01, and *** *p* < 0.001).

**Table 1 jof-07-00135-t001:** Strains List.

Strain	Genotype	Reference
FB1 WT	*a1 b1*	[10]
FB1Δ*b*	*a1 b1 bE/W::hyg^R^*	[11]
FB1Δ*b ump2^Otef^*	*a1 b1 bE/W::hyg^R^ P_otef_ump2, cbx^R^*	This Study
FB2 WT	*a2 b2*	[10]
1/2 WT	*a1 b1*	[14]
2/9 WT	*a2 b2*	[14]
RK1613	*a1 bE1 bW1::hyg^R^*	[15]
RK1725	*a1 bW1 bE1::hyg^R^*	[15]
RK1613 *ump2^Otef^*	*a1 bE1 bW1::hyg^R^ P_otef_ump2, cbx^R^*	This study
RK1725 *ump2^Otef^*	*a1 bW1 bE1::hyg^R^ P_otef_ump2, cbx^R^*	This study

**Table 2 jof-07-00135-t002:** qRT-PCR Primers.

Gene Name	Primers	Sequences (5′→3′)
Guanine nucleotide exchange factor (um04869)	rt-eif-2B-F	CAAATGCGATCCCGAACAG
	rt-eif-2B-R	GGGACACCACTTGTCAAGCA
*ump2* (um05889)	rt-Ump2-F	TGGGTCCCGTTCTCATTTTC
	rt-Ump2-R	AGGCGATGGGATTGTAGACAA
*prf1* (um02713)	rt-Prf1-F	CAGCACCAAGGTGGAAAGGT
	rt-Prf1-R	GAATTGCCACGTGTTTGCAA
*mfa1* (um02382)	rt-Mfa1-F	ATGCTTTCGATCTTCGCTCAG
	rt-Mfa1-R	TAGCCGATGGGAGAACCGT
*bE1* (um00577)	b1E1ft123	GCAACAAAAGATACCCAACGA
	b1E1rt334	TTCGACACCCTACATCAGGAC
*bW1* (um00578)	b1W1ft1559	TCGAGTCTGCCTCAATTCCT
	b1W1rt1827	CTCTCCTATGCTGGCTCCAC
*bE2*	bE2 fw bE2 rv	CAGGGGCAATAGGAAAGTCA ACCATTTTCGACCTCGTCAG
*bW2*	bW2F bW2R	CGTGGAGCCTACGGAATCA GGAGGTGACTCGTGTCTGGAA
*rbf1* (um03172)	RT Rbf1 F	AGGGTGTGGCAAATCGTTCT
	RT Rbf1 R	TCGGCATCAGCATGGTTTC
*pra1* (um02383)	Pra1 qrt F	ACTCGATGGTCTGGTGGAAG
	Pra1 qrt R	CTCACGCTCAATTCGCAATA

## Data Availability

The data presented in this study are available on request from the corresponding author. The data for qRT-PCR are stored and archived on the computer for the ABI StepOne Real-Time PCR instrument, but have not been transferred to a publicly available website.

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
