# Peer review of "Hungry for Sex: Differential Roles for Ustilago maydisb Locus Components in Haploid Cells vis à vis Nutritional Availability"

_jof, 2021, doi:10.3390/jof7020135_

Round 1

Reviewer 1 Report

This MS by Wallen et al. reports different roles for bEast (bE) and bWest (bW) in haploid cells of Ustilago maydis. They showed that these homeodomain proteins were expressed differentially in haploid cells under low ammonium conditions. Although further experiments and investigation is needed to better understand both the role and gene regulation of bE and bW, this manuscript describes several impressive efforts to explore relationship between these proteins and mating response pathway. However, I think that the authors need to substantiate that (or at least discuss whether) conclusion based on experimental data of expression level of proteins are reasonable.

Minor comments:

-It is important to demonstrate that proteins are stably and sufficiently expressed. For example, they emphasized thatoverexpression of ump2 rescue loss of ammonium response phenotypes in haploid cells, not phenotypes related to mating (P.8), I wonder whether ump2 overexpression of this experiment is sufficient for mating response.

-They indicated that FB1ΔbW1 ump2potef show phenotypes with very ‘thick’ filaments, but this interpretation should be quantitatively evaluated.

There are several errors (order of Figures etc.) throughout the manuscript. Some cartoons may be helpful for general readers to better understand this study.

Author Response

This MS by Wallen et al. reports different roles for bEast (bE) and bWest (bW) in haploid cells of Ustilago maydis. They showed that these homeodomain proteins were expressed differentially in haploid cells under low ammonium conditions. Although further experiments and investigation is needed to better understand both the role and gene regulation of bE and bW, this manuscript describes several impressive efforts to explore relationship between these proteins and mating response pathway. However, I think that the authors need to substantiate that (or at least discuss whether) conclusion based on experimental data of expression level of proteins are reasonable.

We very much appreciate the description of our efforts in elucidating these observations. We are now more circumspect about the exact nature of the interactions of the proteins in question (Discussion, p. 14, 1st full para.)

Minor comments:

-It is important to demonstrate that proteins are stably and sufficiently expressed. For example, they emphasized thatoverexpression of ump2 rescue loss of ammonium response phenotypes in haploid cells, not phenotypes related to mating (P.8), I wonder whether ump2 overexpression of this experiment is sufficient for mating response.

In terms of protein stability, the only direct measures of proteins in this study are now provided in S4 Fig., showing that there does not appear to be increased Ump2 protein from the endogenous promoter when ump2 is overexpressed from the ectopic otef promoter. Again, we show in Fig. 3 that ovexpression of ump2 is not sufficient to induce mating in a strain deleted for b locus. This is shown via the failure of the FB1 strain deleted for b and overexpressing ump2 being unable to produce the fuzz reaction alone or in combination with wild type FB2. Moreover, such attempted matings are unable to cause infection of maize plants (not shown).

-They indicated that FB1ΔbW1 ump2potef show phenotypes with very ‘thick’ filaments, but this interpretation should be quantitatively evaluated.

This characterization of the FB1ΔbW1 ump2potef  as having “very thick filaments” has been removed. Now we state that they are qualitatively different and are an exaggeration of the phenotype seen in FB1ΔbW1 (p. 9, para. 2)

There are several errors (order of Figures etc.) throughout the manuscript. Some cartoons may be helpful for general readers to better understand this study.

We have carefully gone through and fixed these. In particular, there had initially been problems with S1 Fig and S2 Fig order in the text. These have now been fixed. As for a cartoon, a relevant cartoon is provided in our previously published work, Paul et al. 2018 Fungal Biology, Fig. 7, that shows a model of the proposed interactions. We do not feel it would be appropriate to re-use that here.

Reviewer 2 Report

The authors demonstrate in their manuscript a role of mating locus components on haploid development. The work is well done with appropriate statistics and experiments.

At some points the manuscript must be improved.

  1. There are many descriptions for high and low ammonium conditions like rich medium, deplete or replete or limited conditions, high and low ammonium or nitrogen. To my opinion these wordings make the story more complicated than it is. I would propose to use one short term like hAm or lAm - Maybe you have a better idea.
  2. Figure 2A and Figure 2B: Please use scale bars of same colour and size. Please organise the images with identical distances and sizes.
  3. The authors used the plasmid p123 to express ump2. Did the authors checked for integration frequency of the SspI linearised plasmid in the ip-locus? They can use the Potef-sequence as probe on EcoRV digested gDNA in a Southern blot. Different integration frequency can be the reason for different expression levels.
  4. Supplementary Figures S2 and S3: Please use identical colours for the columns as in the main manuscript. S2B: increase the size of text and reduce the thickness of the column. S3: itacilize the genes.
  5. It would be interesting if other b-loci behave similar towards b1 and/or b2. Please discuss this point.

Author Response

The authors demonstrate in their manuscript a role of mating locus components on haploid development. The work is well done with appropriate statistics and experiments.

We thank the reviewer for these comments.

At some points the manuscript must be improved.

  1. There are many descriptions for high and low ammonium conditions like rich medium, deplete or replete or limited conditions, high and low ammonium or nitrogen. To my opinion these wordings make the story more complicated than it is. I would propose to use one short term like hAm or lAm - Maybe you have a better idea.

We have changed to High NH4   or Low NH4 throughout the manuscript.

2.Figure 2A and Figure 2B: Please use scale bars of same colour and size. Please organise the images with identical distances and sizes.

We have made all these changes.

3.The authors used the plasmid p123 to express ump2. Did the authors checked for integration frequency of the SspI linearised plasmid in the ip-locus? They can use the Potef-sequence as probe on EcoRV digested gDNA in a Southern blot. Different integration frequency can be the reason for different expression levels.

We did not check for integration frequency/copy number of insertions. However, we did examine several independently isolated mutants, and each showed similar levels of Ump2 expression as those shown in the figures.

4.Supplementary Figures S2 and S3: Please use identical colours for the columns as in the main manuscript. S2B: increase the size of text and reduce the thickness of the column. S3: itacilize the genes.

We have made all these changes.

5.It would be interesting if other b-loci behave similar towards b1 and/or b2. Please discuss this point.

Thank you for this suggestion. We have discussed this idea now in Discussion, page 14, last paragraph: “Here, we observed differences in expression levels for different bE and bW alleles (Fig 1E and F). We do not understand the basis of this finding, but it would be interesting to examine additional b locus alleles for differences in levels of expression in haploids as a response to nitrogen availability.”